# Factors Associated with Significant Platelet Count Improvement in Thrombocytopenic Chronic Hepatitis C Patients Receiving Direct-Acting Antivirals

**DOI:** 10.3390/v14020333

**Published:** 2022-02-07

**Authors:** Yen-Chun Chen, Te-Sheng Chang, Chien-Hung Chen, Pin-Nan Cheng, Ching-Chu Lo, Lein-Ray Mo, Chun-Ting Chen, Chung-Feng Huang, Hsing-Tao Kuo, Yi-Hsiang Huang, Chi-Ming Tai, Cheng-Yuan Peng, Ming-Jong Bair, Ming-Lun Yeh, Chih-Lang Lin, Chun-Yen Lin, Pei-Lun Lee, Lee-Won Chong, Chao-Hung Hung, Jee-Fu Huang, Chi-Chieh Yang, Jui-Ting Hu, Chih-Wen Lin, Chia-Chi Wang, Wei-Wen Su, Tsai-Yuan Hsieh, Chih-Lin Lin, Wei-Lun Tsai, Tzong-Hsi Lee, Guei-Ying Chen, Szu-Jen Wang, Chun-Chao Chang, Sheng-Shun Yang, Wen-Chih Wu, Chia-Sheng Huang, Chou-Kwok Hsiung, Chien-Neng Kao, Pei-Chien Tsai, Chen-Hua Liu, Mei-Hsuan Lee, Chia-Yen Dai, Jia-Horng Kao, Wan-Long Chuang, Han-Chieh Lin, Chi-Yi Chen, Kuo-Chih Tseng, Ming-Lung Yu

**Affiliations:** 1Department of Internal Medicine, Dalin Tzu Chi Hospital, Buddhist Tzu Chi Medical Foundation, Chiayi School of Medicine, Tzuchi University, Hualien 970, Taiwan; bcelltcell@gmail.com; 2Division of Hepatogastroenterology, Department of Internal Medicine, Chiayi Chang Gung Memorial Hospital and College of Medicine, Chang Gung University, Taoyuan 333, Taiwan; cgmh3621@cgmh.org.tw (T.-S.C.); chh4366@yahoo.com.tw (C.-H.H.); 3Division of Hepato-Gastroenterology, Department of Internal Medicine, Kaohsiung Chang Gung Memorial Hospital and Chang Gung University College of Medicine, Taoyuan 333, Taiwan; e580306@ms31.hinet.net; 4Division of Gastroenterology and Hepatology, Department of Internal Medicine, National Cheng Kung University Hospital, College of Medicine, National Cheng Kung University, Tainan 701, Taiwan; pinnancheng@gmail.com; 5Division of Gastroenterology, Department of Internal Medicine, St. Martin De Porres Hospital, Chiayi 600, Taiwan; locc1965@yahoo.com.tw; 6Division of Gastroenterology, Tainan Municipal Hospital, Tainan 701, Taiwan; moleinray@gmail.com; 7Division of Gastroenterology, Department of Internal Medicine Tri-Service General Hospital Penghu Branch, National Defense Medical Center, Taipei 114, Taiwan; gp0921543gp@yahoo.com.tw; 8Division of Gastroenterology, Department of Internal Medicine, Tri-Service General Hospital, National Defense Medical Center, Taipei 114, Taiwan; tyh1216@ms46.hinet.net; 9Hepatobiliary Division, Department of Internal Medicine and Hepatitis Center, Kaohsiung Medical University Hospital, Kaohsiung Medical University, Kaohsiung 807, Taiwan; fengcheerup@gmail.com (C.-F.H.); pctsai1225@gmail.com (P.-C.T.); d820195cy@gmail.com (C.-Y.D.); waloch@kmu.edu.tw (W.-L.C.); 10School of Medicine and Hepatitis Research Center, College of Medicine, Center for Cancer Research, Center for Liquid Biopsy, Cohort Research, Kaohsiung Medical University, Kaohsiung 807, Taiwan; minglunyeh@gmail.com (M.-L.Y.); jf71218@gmail.com (J.-F.H.); 11Division of Gastroenterology and Hepatology, Department of Internal Medicine, Chi Mei Medical Center, Yongkang District, Tainan 710, Taiwan; kuohsingtao@gmail.com; 12Division of Gastroenterology and Hepatology, Department of Internal Medicine, Taipei Veterans General Hospital, Taipei 11217, Taiwan; yhhuang@vghtpe.gov.tw (Y.-H.H.); hclin@vghtpe.gov.tw (H.-C.L.); 13Institute of Clinical Medicine, School of Medicine, National Yang-Ming Chiao Tung University, Taipei 112, Taiwan; 14Department of Internal Medicine, E-Da Hospital, Kaohsiung 824, Taiwan; chimingtai@gmail.com; 15School of Medicine, College of Medicine, I-Shou University, Kaohsiung 824, Taiwan; 16Center for Digestive Medicine, Department of Internal Medicine, China Medical University Hospital, School of Medicine, China Medical University, Taichung 406, Taiwan; cypeng@mail.cmuh.org.tw; 17Division of Gastroenterology, Department of Internal Medicine, Taitung Mackay Memorial Hospital, Taitung 950, Taiwan; a5963@mmh.org.tw; 18Mackay Medical College, New Taipei City 252, Taiwan; 19Department of Internal Medicine, Kaohsiung Municipal Siaogang Hospital, Kaohsiung Medical University Hospital, Kaohsiung Medical University, Kaohsiung 807, Taiwan; 20Department of Gastroenterology and Hepatology, Keelung Chang Gung Memorial Hospital, Keelung 204, Taiwan; wn49792000@yahoo.com.tw; 21Department of Gastroenterology and Hepatology, Chang Gung Memorial Hospital, Linkou Branch, Linkou 333, Taiwan; chunyenlin@gmail.com; 22Graduate Institute of Biomedical Science, College of Medicine, Chang Gung University, Taoyuan 333, Taiwan; 23Chi Mei Medical Center, Liouying Division of Gastroenterology and Hepatology, Department of Internal Medicine, Tainan 736, Taiwan; peilun57@yahoo.com.tw; 24Division of Hepatology and Gastroenterology, Department of Internal Medicine, Shin Kong Wu Ho-Su Memorial Hospital, Taipei 111, Taiwan; M002291@ms.skh.org.tw; 25School of Medicine, Fu-Jen Catholic University, New Taipei City 242, Taiwan; 26Department of Internal Medicine, Kaohsiung Municipal Ta-Tung Hospital, Kaohsiung Medical University Hospital, Kaohsiung Medical University, Kaohsiung 807, Taiwan; 27Department of Gastroenterology, Division of Internal Medicine, Show Chwan Memorial Hospital, Changhua 500, Taiwan; benz2yang@gmail.com; 28Liver Center, Cathay General Hospital, Taipei 106, Taiwan; huyeh0203@gmail.com; 29Division of Gastroenterology and Hepatology, E-Da Dachang Hospital, School of Medicine, College of Medicine, I-Shou University, Kaohsiung 824, Taiwan; lincw66@gmail.com; 30Taipei Tzu Chi Hospital, Buddhist Tzu Chi Medical Foundation and School of Medicine, Tzu Chi University, New Taipei City 231, Taiwan; uld888@yahoo.com.tw; 31Department of Gastroenterology and Hepatology, Changhua Christian Hospital, Changhua 500, Taiwan; vincentsu6195@gmail.com; 32Department of Gastroenterology, Renai Branch, Taipei City Hospital, Taipei 106, Taiwan; dab53@tpech.gov.tw; 33Division of Gastroenterology and Hepatology, Department of Internal Medicine, Kaohsiung Veterans General Hospital, Kaohsiung 813, Taiwan; tsaiwl@yahoo.com.tw; 34Division of Gastroenterology and Hepatology, Far Eastern Memorial Hospital, New Taipei City 220, Taiwan; thleekimo@yahoo.com.tw; 35Penghu Hospital, Ministry of Health and Welfare, Penghu 880, Taiwan; b101090138@gmail.com; 36Division of Gastroenterology, Department of Internal Medicine, Yuan’s General Hospital, Kaohsiung 802, Taiwan; ezpperqoo@gmail.com; 37Division of Gastroenterology and Hepatology, Department of Internal Medicine, Taipei Medical University Hospital, Taipei 110, Taiwan; chunchao@tmu.edu.tw; 38Division of Gastroenterology and Hepatology, Department of Internal Medicine, School of Medicine, College of Medicine, Taipei Medical University, Taipei 110, Taiwan; 39Division of Gastroenterology & Hepatology, Department of Internal Medicine, Taichung Veterans General Hospital, Taichung 407, Taiwan; yansh2525@gmail.com; 40Wu Wen-Chih Clinic, Kaohsiung 830, Taiwan; a0927168092@gmail.com; 41Yang Ming Hospital, Chiayi 600, Taiwan; gigi9735@yahoo.com.tw; 42Chou Kwok Hsiung Clinic, Penghu 880, Taiwan; neil855011@gmail.com; 43National Taiwan University Hospital Hsin-Chu Branch, Hsinchu 300, Taiwan; eagleshone@gmail.com; 44Hepatitis Research Center, Department of Internal Medicine, National Taiwan University Hospital, Taipei 106, Taiwan; jacque_liu@mail2000.com.tw (C.-H.L.); kaojh@ntu.edu.tw (J.-H.K.); 45Institute of Clinical Medicine, National Yang-Ming Chiao-Tung University, Taipei 112, Taiwan; meihlee@ntu.edu.tw; 46Division of Gastroenterology and Hepatology, Department of Medicine, Ditmanson Medical Foundation Chiayi Christian Hospital, Chiayi 600, Taiwan; 5137ccy@gmail.com; 47Institute of Biomedical Sciences, National Sun Yat-sen University, Kaohsiung 804, Taiwan; 48National Pingtung University of Science and Technology, Pingtung 912, Taiwan

**Keywords:** hepatitis C virus, chronic hepatitis C, direct-acting antivirals, platelet count, thrombocytopenia, significant platelet count improvement, sustained virologic response

## Abstract

To clarify the predictive factors of significant platelet count improvement in thrombocytopenic chronic hepatitis C (CHC) patients. CHC patients with baseline platelet counts of <150 × 10^3^/μL receiving direct-acting antiviral (DAA) therapy with at least 12-weeks post-treatment follow-up (PTW12) were enrolled. Significant platelet count improvement was defined as a ≥10% increase in platelet counts at PTW12 from baseline. Platelet count evolution at treatment week 4, end-of-treatment, PTW12, and PTW48 was evaluated. This study included 4922 patients. Sustained virologic response after 12 weeks post-treatment was achieved in 98.7% of patients. Platelet counts from baseline, treatment week 4, and end-of-treatment to PTW12 were 108.8 ± 30.2, 121.9 ± 41.1, 123.1 ± 43.0, and 121.1 ± 40.8 × 10^3^/μL, respectively. Overall, 2230 patients (45.3%) showed significant platelet count improvement. Multivariable analysis revealed that age (odds ratio (OR) = 0.99, 95% confidence interval (CI): 0.99–1.00, *p* = 0.01), diabetes mellitus (DM) (OR = 1.20, 95% CI: 1.06–1.38, *p* = 0.007), cirrhosis (OR = 0.66, 95% CI: 0.58–0.75, *p* < 0.0001), baseline platelet counts (OR = 0.99, 95% CI: 0.98–0.99, *p* < 0.0001), and baseline total bilirubin level (OR = 0.80, 95% CI: 0.71–0.91, *p* = 0.0003) were independent predictive factors of significant platelet count improvement. Subgroup analyses showed that patients with significant platelet count improvement and sustained virologic responses, regardless of advanced fibrosis, had a significant increase in platelet counts from baseline to treatment week 4, end-of-treatment, PTW12, and PTW48. Young age, presence of DM, absence of cirrhosis, reduced baseline platelet counts, and reduced baseline total bilirubin levels were associated with significant platelet count improvement after DAA therapy in thrombocytopenic CHC patients.

## 1. Introduction

Thrombocytopenia is a common manifestation of chronic hepatitis C (CHC) infection, with a prevalence of 0.16–45.4% in CHC patients. CHC patients with advanced fibrosis or cirrhosis may have severe thrombocytopenia and high rates of thrombocytopenia [1,2]. Increased platelet destruction and decreased platelet production cause thrombocytopenia. The pathophysiology of thrombocytopenia includes splenomegaly-induced hypersplenism, autoimmune responses such as platelet-associated immunoglobulin G against platelet surface antigen, inadequate thrombopoietin (TPO) production due to advanced liver fibrosis, and possible bone marrow suppression by hepatitis C virus (HCV). In the era of interferon (IFN)-based therapy for CHC, achieving a sustained virologic response (SVR) improves platelet counts in patients with advanced liver fibrosis after a long-term follow-up period [3,4]. In patients without SVR, platelet counts decrease [4]. Improvement in platelet counts after a long-term follow-up period is most likely due to the improved liver fibrosis and portal hypertension, further decreasing the risk of hepatocellular carcinoma (HCC) [4,5,6]. IFN-free direct-acting antivirals (DAAs) have been the standard treatment for CHC [7,8,9,10]. Platelet count improvement has also been reported, and could be observed within a short-term follow-up period [11,12,13,14,15]. Liver fibrosis status is less likely to improve rapidly and significantly immediately after initiation of DAA therapy [16]. Rapid or early platelet count improvement may be associated with other factors, such as the aforementioned viral effect [17] or autoimmune responses [11]. In thrombocytopenic CHC patients who will have an elective invasive procedure, DAA treatment might be helpful to improve platelet counts in addition to platelet transfusion alone. Our previous study that enrolled CHC patients receiving DAA therapy revealed that patients without significant platelet count improvement would have more advanced hepatic fibrosis and more liver-related complications such as HCC, splenomegaly, and ascites [15]. However, the sample size of that study was relatively small. After the long-term follow-up period, it remains unclear what pattern the platelet count evolution would follow in these DAA-treated CHC patients. Therefore, we conducted this large-scale study to clarify the predictive factors of significant platelet count improvement and the patterns of platelet count evolution.

## 2. Materials and Methods

### 2.1. Patient Selection

This retrospective study enrolled patients from the Taiwan Association for the Study of Liver (TASL) HCV Registry (TACR). The TACR is a nationwide registry program organized by the TASL, which sets up and manages the database of patients with HCV who receive DAA therapy in Taiwan. From May 2017 to August 2020, 38 study centers were enrolled in the registry. Individual patient records were reviewed, and data were extracted and validated at each participating study center using a standardized case report form and a unified coding dictionary. Inclusion criteria: (1) patients aged >20 years, (2) patients with positive anti-hepatitis C antibody and detectable HCV RNA levels, (3) patients with baseline thrombocytopenia (platelet counts <150 × 10^3^ /μL) [1], and (4) patients with a complete DAA course with at least 12 weeks of post-treatment follow-up (PTW12) after end-of-treatment (EOT). Patients with human immunodeficiency virus coinfection and concurrent cancer (including HCC) receiving treatment were excluded. The treatment duration and regimens were based on the regulations of the Health and Welfare Department of Taiwan and the guidelines [7,8,18]. Among the 25,905 CHC patients registered in the TACR, 4922 patients who fulfilled the inclusion criteria were enrolled in our study (Figure 1). This study was approved by the Research Ethics Committee of each participating center.

### 2.2. Clinical and Laboratory Monitoring

Baseline characteristics, including age, sex, hypertension, diabetes mellitus (DM), cirrhosis, decompensated cirrhosis, history of variceal bleeding, chronic hepatitis B virus (HBV) coinfection, ascites, history of hepatic encephalopathy, IFN-experienced condition, and chronic kidney disease (CKD), were recorded using chart review. Liver cirrhosis was defined based on any of the following modalities: liver histology; transient elastography (FibroScan^®^; Echosens, Paris, France) (>12 kPa); acoustic radiation force impulse (>1.98 m/s); fibrosis-4 (FIB-4) index (score > 6.5); or the presence of clinical, radiological, endoscopic, or laboratory evidence of cirrhosis and/or portal hypertension [18]. Decompensated cirrhosis was defined as the deterioration of liver function in patients with cirrhosis and was characterized by jaundice (total bilirubin levels >2 mg/dL), variceal bleeding, ascites, or hepatic encephalopathy [19]. HCC was diagnosed either by biopsy or imaging in the setting of liver cirrhosis [20]. SVR was defined as undetectable HCV RNA levels 12 weeks after EOT (PTW12) [21].

Laboratory data (serum platelet, aspartate aminotransferase (AST), alanine aminotransferase (ALT), albumin, total bilirubin, prothrombin time with international normalized ratio (PT INR), and creatinine) were assessed at the hepatogastrointestinal outpatient clinic at baseline, EOT, PTW12, and 48 weeks after EOT (PTW48). Alpha-fetoprotein (AFP) level examination and abdominal ultrasonography were performed at baseline and 3–6 months after initiation of DAA therapy. HCV RNA was quantified at baseline, EOT, and PTW12. Due to variations in platelet counts at different time points of blood tests, we defined significant platelet count improvement as a ≥10% increase in platelet counts from baseline to PTW12 ((platelet counts at PTW12−platelet counts at baseline)/platelet counts at baseline) [13].

### 2.3. Statistical and Data Analysis

The commercial statistical software package (SAS version 9.4; SPSS, version 20) was used for all statistical analyses. Frequency was compared between groups with and without significant platelet count improvement using the X^2^ test with Yate’s correction or Fisher’s exact test. Group mean values (presented as mean ± standard deviation (SD)) were compared using Student’s t-test or the nonparametric Mann–Whitney *U*-test when appropriate. Stepwise logistic regression analysis using the bidirectional elimination method was performed to determine factors associated with significant platelet count improvement by analyzing the covariates in the single variable analysis (*p* < 0.1), with gender adjusted at the same time. A collinear test was used to determine whether the independent factors were highly correlated. All statistical analyses were based on two-sided hypothesis tests with a significance level of *p* < 0.05.

## 3. Results

### 3.1. Baseline Characteristics of CHC Patients

A total of 4922 patients who completed their follow-up at PTW12 were included in the final analysis. The baseline characteristics are summarized in Table 1. Overall, 2030 patients (41.2%) were men, and the mean age of patients was 65.0 ± 10.7 years. In this cohort, 1245 patients had DM (25.3%), 2051 had cirrhosis (41.7%), 270 had decompensated cirrhosis (5.5%), 361 had HBV coinfection (7.3%), and 4859 achieved SVR12 (98.7%).

### 3.2. Platelet Count Evolution from Baseline to 48 Weeks after EOT (PTW48)

As shown in Figure 2a, the platelet counts from baseline to treatment week 4 (W4), EOT, PTW12, and PTW48 were 108.6 ± 30.1, 122.4 ± 41.0, 123.2 ± 43.0, 120.6 ± 40.2, and 124.4 ± 45.0 × 10^3^/μL, respectively (*p* < 0.001 for W4 vs. baseline, EOT vs. baseline, PTW12 vs. baseline, and PTW48 vs. baseline). An increase in platelet counts was noted during the treatment and follow-up periods. We performed a subgroup analysis to investigate further whether platelet count improvement occurred before W4. Among the studied cohort, 243 patients had their platelet counts evaluated at W1, W2, and W4. Platelet counts at baseline, W1, W2, and W4 were 106.5 ± 30.7, 119.3 ± 37.2, 117.9 ± 36.7, and 121.4 ± 38.5 × 10^3^/μL, respectively (*p* < 0.001 for W1 vs. baseline, W2 vs. baseline, and W4 vs. baseline). An increase in platelet counts occurred as early as week one after the initiation of DAA therapy.

### 3.3. Factors Associated with Significant Platelet Count Improvement

Significant platelet count improvement was observed in 2230 (45.3%) patients with thrombocytopenic CHC who were receiving DAA therapy. Table 1 shows that between patients with and without significant platelet count improvement, age (64.6 ± 10.7 vs. 65.3 ± 10.6 years), DM (*n* = 598 (26.8%) vs. *n* = 647 (24.0%)), cirrhosis (*n* = 849 (38.1%) vs. *n* = 1202 (44.7%)), decompensated cirrhosis (*n* = 100 (4.5%) vs. *n* = 170 (6.3%)), ascites (*n* = 55 (2.5%) vs. *n* = 104 (3.9%)), baseline platelet counts (105.7 ± 31.3 vs. 111.4 ± 28.6 × 10^3^/μL), baseline total bilirubin level (0.96 ± 0.52 vs. 1.01 ± 0.56 mg/dL), and baseline creatinine level (1.24 ± 1.78 vs. 1.13 ± 1.47 mg/dL) were associated with significant platelet count improvement after single variable analysis. Multivariable analysis (Table 2) revealed that age (odds ratio (OR) = 0.99, 95% confidence interval (CI): 0.99–1.00, *p* = 0.01), DM (OR = 1.20, 95% CI: 1.06–1.38, *p* = 0.007), cirrhosis (OR = 0.66, 95% CI: 0.58–0.75, *p* < 0.0001), baseline platelet counts (OR = 0.99, 95% CI: 0.98–0.99, *p* < 0.0001), and baseline total bilirubin level (OR = 0.80, 95% CI: 0.71–0.91, *p* = 0.0003) were independent predictive factors of significant platelet count improvement, indicating that younger age, presence of DM, absence of cirrhosis, reduced baseline platelet counts, or reduced baseline total bilirubin level are associated with significant platelet count improvement after DAA therapy.

### 3.4. Platelet Count Evolution in Patients with or without Significant Platelet Count Improvement

Platelet count evolution results based on significant platelet count improvement are presented in Figure 2b. In patients with significant platelet count improvement, the platelet counts at baseline, W4, EOT, PTW12, and PTW48 were 105.6 ± 31.2, 128.2 ± 42.8, 131.6 ± 43.7, 138.0 ± 42.7, and 134.0 ± 45.1 × 10^3^/µL, respectively (*p* < 0.001 for W4 vs. baseline, EOT vs. baseline, PTW12 vs. baseline, and PTW48 vs. baseline). In patients without significant platelet count improvement, platelet counts at baseline, W4, EOT, PTW12, and PTW48 were 111.2 ± 28.9, 117.8 ± 38.92, 116.4 ± 41.2, 106.4 ± 31.7, and 117.3 ± 43.7 × 10^3^/µL, respectively (*p* < 0.001 for W4 vs. baseline, EOT vs. baseline, PTW12 vs. baseline, and PTW48 vs. baseline). Furthermore, platelet counts between patients with and without significant platelet count improvement were significantly different at baseline (*p* < 0.001), W4 (*p* < 0.001), EOT (*p* < 0.001), PTW12 (*p* < 0.001), and PTW48 (*p* < 0.001).

### 3.5. Platelet Count Evolution Based on DAA Treatment Response

Only 45 patients (1.2%) with platelet counts examined at EOT and PTW12 did not achieve SVR. Platelet count evolution based on the DAA treatment response is reported in Figure 3. Platelet counts in patients without SVR at baseline, W4, EOT, PTW12, and PTW48 were 95.0 ± 31.8, 108.7 ± 38.9, 110.2 ± 36.9, 103.1 ± 35.4, and 100.4 ± 34.8 × 10^3^/µL, respectively (W4 vs. baseline, *p* < 0.05; EOT vs. baseline, *p* = 0.003; PTW12 vs. baseline, *p* = 0.115; and PTW48 vs. baseline, *p* = 0.290). After DAA treatment, non-SVR patients showed a significant increase in platelet counts until EOT, but this increase became insignificant at and after PTW12. In contrast, platelet counts in SVR patients at baseline, W4, EOT, PTW12, and PTW48 were 108.8 ± 30.1, 122.6 ± 41.1, 123.4 ± 43.0, 120.9 ± 40.2, and 124.9 ± 45.1 × 10^3^/μL, respectively (*p* < 0.001 for W4 vs. baseline, EOT vs. baseline, PTW12 vs. baseline, and PTW48 vs. baseline). Furthermore, platelet counts between non-SVR and SVR patients were significantly different at baseline (*p* = 0.006), W4 (*p* = 0.030), EOT (*p* = 0.021), PTW12 (*p* = 0.002), and PTW48 (*p* = 0.004).

### 3.6. Platelet Count Evolution Based on Baseline Fibrosis Status

A total of 2782 (76.2%) patients with platelet counts examined at EOT and PTW12 had advanced fibrosis according to FIB-4 scores (cutoff value, 3.25). Platelet count evolution results based on fibrosis status are shown in Figure 4. In patients with non-advanced fibrosis, platelet counts at baseline, W4, EOT, PTW12, and PTW48 were 131.1 ± 15.5, 145.1 ± 29.7, 145.0 ± 29.5, 142.9 ± 26.9, and 151.4 ± 37.9 × 10^3^/μL, respectively (*p* < 0.001 for W4 vs. baseline, EOT vs. baseline, PTW12 vs. baseline, and PTW48 vs. baseline). In patients with advanced fibrosis, platelet counts at baseline, W4, EOT, PTW12, and PTW48 were 101.7 ± 30.1, 115.4 ± 41.4, 116.4 ± 44.1, 113.7 ± 41.2, and 117.8 ± 44.3 × 10^3^/μL, respectively (*p* < 0.001 for W4 vs. baseline, EOT vs. baseline, PTW12 vs. baseline, and PTW48 vs. baseline). Patients showed a significant increase in platelet counts at W4, EOT, PTW12, and PTW48 compared with those at baseline, regardless of the fibrosis status. Furthermore, platelet counts between the non-advanced and advanced fibrosis groups were significantly different at baseline (*p* < 0.001), W4 (*p* < 0.001), EOT (*p* < 0.001), PTW12 (*p* < 0.001), and PTW48 (*p* < 0.001).

## 4. Discussion

In this study, overall platelet counts in thrombocytopenic CHC patients improved after DAA therapy. The predictive factors of significant platelet count improvement included younger age, presence of DM, absence of cirrhosis, reduced baseline platelet counts, and reduced baseline total bilirubin levels. Besides, significantly improved platelet counts are noted as early as one week after the initiation of DAA therapy. Subgroup analysis revealed that patients with significant platelet count improvement would consistently improve from baseline to PTW48. SVR patients showed a significant increase in platelet counts from baseline to PTW48. However, non-SVR patients showed a significant increase until EOT, and the changes became insignificant at and after PTW12 compared with the baseline. Patients would have a significant increase in platelet counts from baseline to PTW48 after DAA therapy, regardless of the fibrosis status.

Platelet counts can improve immediately after DAA treatment in CHC patients [11,12,13,14,15]. Some of these studies did not investigate the predictive factors of significant platelet count improvement. Besides, our previous studies only showed that the fatty liver, baseline platelet counts, and baseline thrombopoietin level were related to significant platelet count improvement [13,15]. In this larger-scale study, we found that age, baseline platelet counts, baseline total bilirubin levels, and cirrhosis were negatively correlated, but the presence of DM was positively correlated with significant platelet count improvement (Table 2). Several mechanisms may account for these improvements after HCV eradication. Platelet count improvement is more likely to be related to a decrease in HCV-induced platelet-associated immunoglobulin G levels [11], improvement of hypersplenism [14], and resolution of HCV-related myelosuppression [14,22]. Older age, higher total bilirubin levels, and the presence of cirrhosis at baseline are associated with more severe liver fibrosis, which is related to the poorer liver capacity to produce TPO [23], and splenomegaly may be more prevalent in them [24]. Reduced serum TPO levels and splenomegaly-related hypersplenism could partly account for the absence of significant platelet count improvement even after DAA therapy. DM was positively associated with significant platelet count improvement. Metabolic syndrome and insulin resistance are associated with elevated platelet counts [25,26], and may partly account for the role of DM in platelet count improvement. This does not mean metabolic syndrome is beneficial because it involves chronic inflammation state and cardiovascular disease [27]. Our previous study found that fatty liver is associated with significant platelet count improvement [13]. Fatty liver is related to metabolic syndrome [28]. However, a study showed no correlation between hyperglycemia and platelet count [29]. In the present study, fatty liver could not be known from primary data. Therefore, we could not evaluate the relationship between DM and fatty liver. Hence, further studies are required to explore the underlying mechanism.

Platelet count improvement was observed at W4 (Figure 2a). After subgroup analysis, we further found that a significant increase in platelet counts would begin just one week after initiation of DAA treatment. HCV RNA levels decreased significantly from baseline to W4 (log HCV RNA, baseline 5.90 ± 0.97 IU/mL, W4 1.23 ± 0.27 IU/mL, *p* < 0.05), which is compatible with previous studies [30,31,32]. Besides, HCV RNA levels significantly decreased within 1–2 weeks after DAA therapy initiation and were even undetectable at EOT, independent of the presence or absence of SVR12 [32]. Hence, this phenomenon suggests that the HCV-induced decrease in platelet counts is ameliorated within several weeks of HCV eradication by DAAs. To determine whether significant platelet count improvement could be observed after a long-term follow-up period (at PTW48), we analyzed changes in platelet counts from baseline to PTW48 between patients with and without significant platelet count improvement (Figure 2b). We found that platelet counts in both subgroups increased before EOT, but this increase only continued in patients with significant platelet count improvement. This indicates that significant platelet count improvement (at PTW12) predicts a similar improvement at PTW48. In patients without significant platelet count improvement, platelet counts decreased at PTW12 through an unknown mechanism, but increased again at PTW48 compared with baseline levels. We also demonstrated the importance of SVR in patients with thrombocytopenic CHC (Figure 3). Patients who achieved SVR had a significant increase in platelet counts from baseline to PTW48, whereas the platelet counts of those without SVR increased significantly at EOT, but the change was insignificant at PTW12 and PTW48 compared with baseline levels. This phenomenon may be explained by the reappearance of HCV after EOT in patients without SVR.

Non-advanced fibrosis patients (FIB-4 score < 3.25) had significantly higher platelet counts at baseline, W4, EOT, PTW12, and PTW48 than advanced fibrosis patients (FIB-4 score ≥ 3.25). Still, both these subgroups would have significantly improved platelet counts at time points after initiation of DAA treatment. This finding is similar to that reported by another study [12], wherein significantly increased platelet counts were noted in CHC patients with HCV elimination, regardless of the presence of cirrhosis. This phenomenon reminds us not to interpret early platelet count improvement as a short-term improvement in liver fibrosis.

This study has some limitations. Using large-scale data of the TACR, two important factors—fatty liver and splenomegaly—could not be analyzed. However, this big data still provided some important results. For example, total bilirubin was negatively associated with significant platelet count improvement in this study. Still, this association was not observed in our previous study, possibly due to the small sample size [13]. Second, a serial evaluation of spleen size, the severity of liver fibrosis, or TPO levels were not performed in many patients in our study. Hence, we are unsure whether platelet count improvement at long-term follow-up time points is related to fibrosis improvement alone. Third, platelet counts improved progressively from baseline to PTW12 and PTW48. Whether such progress reaches a plateau after PTW12 should be clarified in future studies with an even longer-term follow-up period (such as data at PTW96 and PTW144). Fourth, we did not check all patients’ resistance-associated substitutions (RAS). RAS is significantly associated with advanced fibrosis/cirrhosis [33]. Cirrhosis is a negative factor predicting significant platelet count improvement in our study. Therefore, we could not know whether there exists any correlation between them or not.

## 5. Conclusions

Thrombocytopenic CHC patients treated with DAAs generally had a significant increase in platelet counts. Younger age, presence of DM, absence of cirrhosis, and reduced baseline platelet counts and total bilirubin levels can predict significant platelet count improvement. Patients with significant platelet count improvement show persistent improvement at PTW48. The absence of SVR negatively affected platelet count improvement. Regardless of liver fibrosis, patients would have improved platelet counts after DAA treatment.

## Figures and Tables

**Figure 1 viruses-14-00333-f001:**
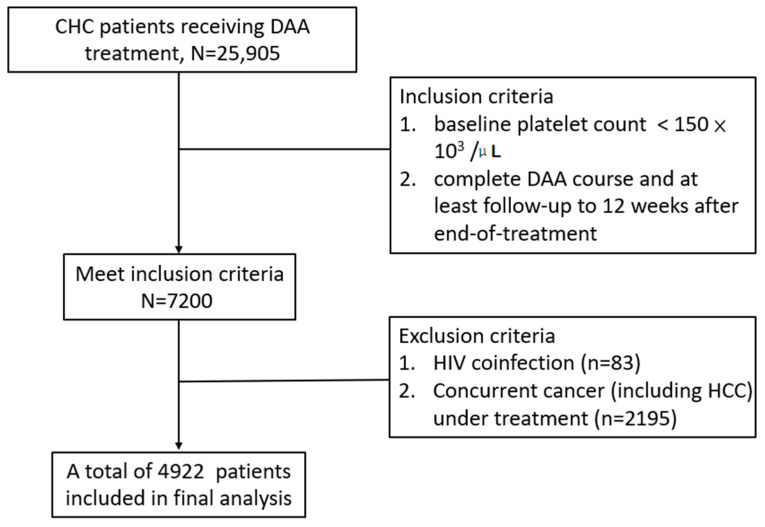
Study flow chart.

**Figure 2 viruses-14-00333-f002:**
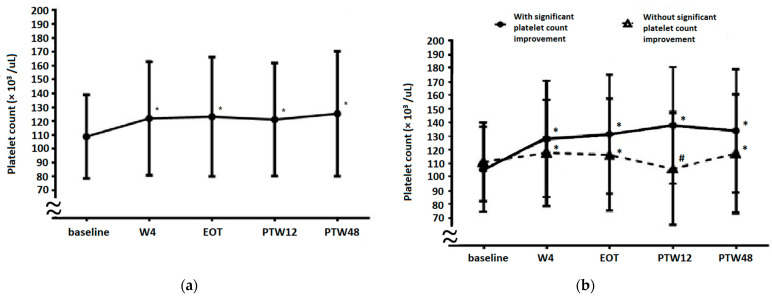
The dynamic changes in platelet counts from baseline to W4, EOT, PTW12, and PTW48 in (**a**) the overall cohort and (**b**) patients with and without significant platelet count improvement. * Platelet counts are significantly increased when compared with baseline levels. # Platelet counts are significantly decreased compared with baseline levels (*p* < 0.001 for W4 vs. baseline, EOT vs. baseline, PTW12 vs. baseline, and PTW48 vs. baseline). W4, week 4; EOT, end-of-treatment; PTW12, 12 weeks post-treatment follow-up; PTW48, 48 weeks post-treatment follow-up.

**Figure 3 viruses-14-00333-f003:**
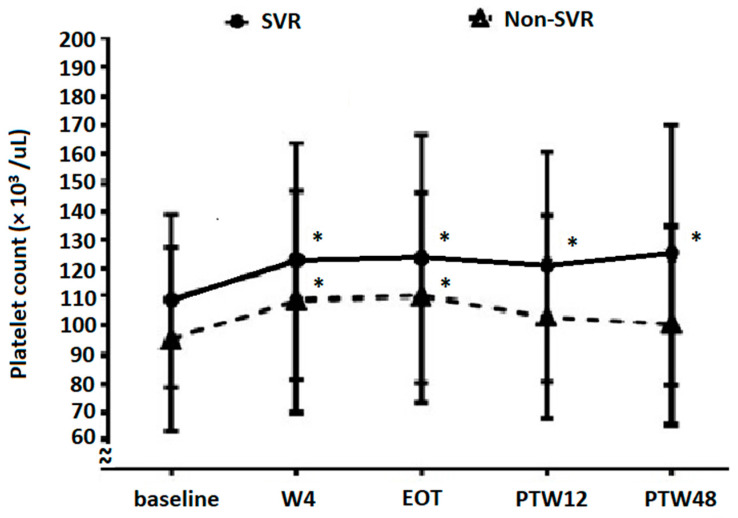
The evolution of platelet counts in patients with SVR12 and without SVR12. * Platelet counts are significantly increased compared with baseline levels (in SVR patients, *p* < 0.001 for W4 vs. baseline, EOT vs. baseline, PTW12 vs. baseline, and PTW48 vs. baseline; in non-SVR patients, W4 vs. baseline, *p* < 0.05; EOT vs. baseline, *p* = 0.003; PTW12 vs. baseline, *p* = 0.115; and PTW48 vs. baseline, *p* = 0.290). W4, week 4; EOT, end-of-treatment; PTW12, 12 weeks post-treatment follow-up; PTW48, 48 weeks post-treatment follow-up.

**Figure 4 viruses-14-00333-f004:**
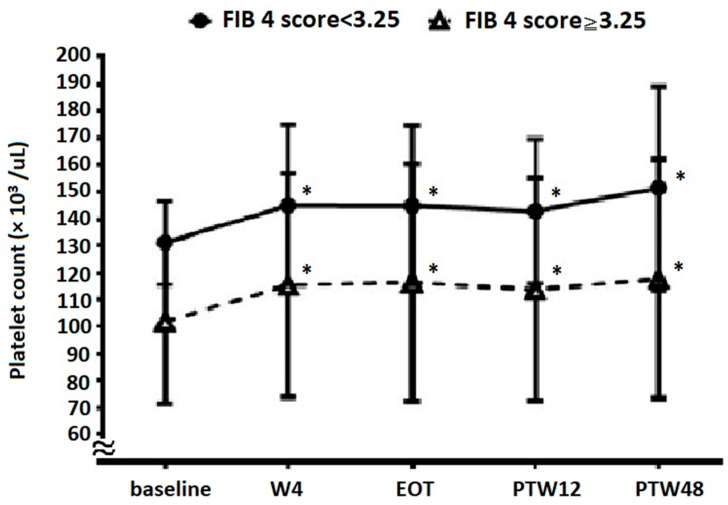
The evolution of platelet count in patients with and without advanced liver fibrosis. * Platelet counts are significantly increased compared with baseline levels (*p* < 0.001 for W4 vs. baseline, EOT vs. baseline, PTW12 vs. baseline, and PTW48 vs. baseline). W4, week 4; EOT, end-of-treatment; PTW12, 12 weeks post-treatment follow-up; PTW48, 48 weeks post-treatment follow-up.

**Table 1 viruses-14-00333-t001:** Baseline characteristics of the thrombocytopenic HCV patients with or without significant platelet count improvement after DAA therapy.

Variables Mean ± SD or *n* (%)	Total (*n* = 4922)	Patients with Significant Platelet Count Improvement ^#^ (*n* = 2230)	Patients without Significant Platelet Count Improvement ^#^ (*n* = 2692)	*p* Value
Baseline clinical characteristics				
Age (years)	65.0 ± 10.7	64.6 ± 10.7	65.3 ± 10.6	0.023
Male sex, *n* (%)	2030 (41.2)	941 (42.2)	1089 (40.5)	0.216
HTN, *n* (%)	3185 (64.7)	1444 (64.8)	1741 (64.7)	0.953
DM, *n* (%)	1245 (25.3)	598 (26.8)	647 (24.0)	0.025
Cirrhosis, *n* (%)	2051 (41.7)	849 (38.1)	1202 (44.7)	<0.0001
Decompensated cirrhosis, *n* (%)	270 (5.5)	100 (4.5)	170 (6.3)	0.005
History of variceal bleeding, *n* (%)	92 (1.9)	33 (1.5)	59 (2.2)	0.067
HBV coinfection, *n* (%)	361 (7.3)	164 (7.3)	197 (7.3)	0.961
Ascites, *n* (%)	159 (3.2)	55 (2.5)	104 (3.9)	0.006
History of HE, *n* (%)	10 (0.2)	6 (0.3)	4 (0.2)	0.350
Treatment experienced, *n* (%)	1167 (23.7)	519 (23.3)	648 (24.1)	0.512
CKD, *n* (%)	377 (7.7)	179 (8.0)	198 (7.4)	0.378
SVR12, *n* (%)	4859 (98.7)	2205 (98.9)	2654 (98.6)	0.367
Baseline laboratory data				
Platelet count (10^3^/μL)	108.8 ± 30.0	105.7 ± 31.3	111.4 ± 28.6	<0.0001
AST (U/L)	78.4 ± 57.6	78.9 ± 54.8	78.1 ± 59.9	0.615
ALT (U/L)	88.7 ± 74.1	90.1 ± 70.3	87.5 ± 77.2	0.214
Total bilirubin (mg/dL)	0.99 ± 0.54	0.96 ± 0.52	1.01 ± 0.56	0.003
Albumin (g/dL)	4.04 ± 0.45	4.05 ± 0.44	4.03 ± 0.46	0.299
PT INR	1.09 ± 0.34	1.09 ± 0.38	1.09 ± 0.31	0.807
Creatinine (mg/dL)	1.18 ± 1.62	1.24 ± 1.78	1.13 ± 1.47	0.025
AFP (ng/mL)	19.4 ± 114.2	21.4 ± 127.3	17.7 ± 102.0	0.272

^#^ Significant platelet count improvement is defined as a ≥10% increase in platelet counts post-treatment week 12 from baseline. HCV, hepatitis C virus; DAA, direct-acting antiviral agent; HTN, hypertension; DM, diabetes mellitus; HBV, hepatitis B virus; HE, hepatic encephalopathy; CKD, chronic kidney disease; SVR, sustained virologic response; ALT, alanine aminotransferase; AST, aspartate aminotransferase; PT, prothrombin time; INR, international normalized ratio; AFP, alpha-fetoprotein.

**Table 2 viruses-14-00333-t002:** Single variable and multivariable analyses associated with significant platelet count improvement in thrombocytopenic HCV patients after DAA therapy.

Variable	Single Variable Analysis		Multivariable Analysis	
	Odds Ratio (95% CI)	*p* Value	Odds Ratio (95% CI)	*p* Value
Baseline clinical characteristics				
Age	0.99 (0.99, 1.00)	0.023	0.99 (0.99, 1.00)	0.01
Male sex	1.07 (0.96, 1.20)	0.216	1.09 (0.72,2.15)	0.171
HTN	1.00 (0.89, 1.12)	0.953		
DM	1.16 (1.01, 1.32)	0.026	1.20 (1.06, 1.38)	0.007
Cirrhosis	0.76 (0.68, 0.85)	<0.0001	0.66 (0.58, 0.75)	<0.0001
Decompensated cirrhosis	0.70 (0.54, 0.90)	0.005	0.99 (0.62, 1.60)	0.992
History of variceal bleeding	0.67 (0.44, 1.03)	0.068	0.80 (0.46,1.39)	0.438
HBV coinfection	1.01 (0.81, 1.25)	0.961		
Ascites	0.63 (0.45, 0.88)	0.006	0.65 (0.38, 1.10)	0.107
History of HE	1.81 (0.51, 6.43)	0.357		
Treatment experienced	0.96 (0.84, 1.09)	0.512		
With CKD	1.10 (0.89, 1.36)	0.378		
With SVR12	1.26 (0.76, 2.10)	0.368		
Baseline laboratory data				
Platelet count	0.41 (0.31, 0.53)	<0.0001	0.99 (0.98, 0.99)	<0.0001
Total bilirubin	0.85 (0.76, 0.95)	0.002	0.80 (0.71, 0.91)	0.0003
Albumin	1.07 (0.94, 1.21)	0.300		
PT INR	1.02 (0.87, 1.21)	0.804		
Creatinine	1.04 (1.00, 1.08)	0.026	1.02 (0.98, 1.06)	0.261
AFP	1.00 (0.99, 1.00)	0.261		

*p* value < 0.1 on the single variable analysis model was analyzed on the multivariable analysis model. HCV, hepatitis C virus; DAA, direct-acting antiviral agent; HTN, hypertension; DM, diabetes mellitus; HE, hepatic encephalopathy; CKD, chronic kidney disease; SVR, sustained virologic response; PT, prothrombin time; INR, international normalized ratio; AFP, alpha-fetoprotein.

## Data Availability

All the data were obtained upon the request to the corresponding authors.

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
