# Peer review of "Factors Associated with Significant Platelet Count Improvement in Thrombocytopenic Chronic Hepatitis C Patients Receiving Direct-Acting Antivirals"

_viruses, 2022, doi:10.3390/v14020333_

Round 1

Reviewer 1 Report

Reviewer(s)' Comments to Author:

In the manuscript entitled “Factors associated with significant platelet count improvement 2 in thrombocytopenic chronic hepatitis C patients receiving direct-acting antivirals” by Yen-Chun Chen et al, the authors investigated predictive factors of significant platelet count improvement in thrombocytopenic chronic hepatitis C patients receiving direct-acting antivirals.

Overall, this manuscript provided evidence that younger age, presence of DM, absence of cirrhosis and ascites, and reduced baseline platelet counts and total bilirubin levels are independent predictive factors for significant platelet count improvement. However, some revisions are worth noting.

Abstract

Line 103-line 106 It is suggested that attaching the p-value after the confidence interval, making the results more complete.

Introduction

Line 128 and line 132 Many places in the text use the “platelet counts”, while other places use the “platelet count”, please check and unify the usage.

Line 133-line 135 Considering that the title of this study underscores the importance of factors associated with significant platelet count improvement, I suggest the authors add more information about the significance and clinical applications of this.

Line 136-line 138 The consistency of previous studies with the present study can be further detailed in the discussion.

Materials and Methods

Line 155 There are mistakes in the spelling of the microliter in the full text. Please replace uL with μL throughout.

Line 176 Please check the consistency of the tenses in the grammar.

Line 198 In Table 2 you have given the results of the single variable and multivariable logistic regression. You say in your methods that variables with p<0.05 in the single variable analysis were included in the multivariable analysis. Did you do a backwards or forwards stepwise procedure to end up with the variables in Table 2? I would suggest adding variables with p<0.1 or 0.2 in the single variable analysis in the multivariable analysis, this will prevent the omission of potentially valuable variables such as the history of variceal bleeding.

Results

Line 243 In Table 1, DM's percentages have an extra % in the bracket while others do not, please standardize the format.

Line 250 Can you replace univariate with single variable and multivariate with multivariable throughout? 

Line 251 In Table 2, the authors have provided several independent risk factors for predicting significant platelet count improvement in thrombocytopenic HCV patients after DAA therapy, have you adjusted it for gender and other important comorbidities? If not, I suggest the authors add this part and it will become a better model with a crude odds ratio and adjusted odds ratio.

Line 225-line 242 In addition, to verify the validity potential of the combined model consisting of these several promising indicators, the classification accuracy of significant platelet count improvement should be further tested with ROC analysis.

Line 265-line 267 The relevant p-values should be presented and noted in the result graphs, same as line 280-line 282 and line 298-line 301.

Discussion

Line 321-line 323 Table 2 The results in Table 2 provided some indication of the independent predictors, but did not provide a good illustration of the correlations. The specific correlation results need to be supplemented by correlation analysis between these indicators and the significant platelet count improvement using Spearman or Pearson correlation analysis, as appropriate.

Line 325 Are there indicators in other studies that suggest significant improvement in platelets with DAA treatment? If so, please cite the relevant literature for additional clarification.

Line 332-line 336 The discussion is deficient. Although the multivariable regression showed an association between platelet count improvement and diabetes, it does not mean that it is beneficial. Hyperglycemia itself may cause platelet elevation, which should be explained in depth in the discussion from the perspective of the mechanism.

Others

The article needs to be checked and edited for some language errors.

Reviewer 2 Report

Dear Authors, 

I've read with pleasure your paper. I think It would be interesting if you can better argue the role of diabetes in platelet count improvement (334-336).

Reviewer 3 Report

The paper is well written and very interesting. The authors studied on a large cohort of patients, factors predicting improvement in platelet counts under DAA treatmnets: the analysis is well performed and results are convincing . The usefulness of such information in the clinical setting might be relevant. As a matter of fact, I believe that this paper deserves publication..

Only two minor criticisms I have:

  1. I think that the results cannot be considered a demonstration, through  indirect, of HCV related myelosuppression, Thus better to remove this sentence from the discussion.
  2. a limitation if the study, that should just be mentioned , is that no data are presented on resistance associated mutations in HCV genome in the cohort: RAS are, in fact, in turn related to fibrosis and other co-factors relevant in clinical course and response to treatment (Di Stefano M et al.  New Microbiologica 44 2021).

Round 2

Reviewer 1 Report

After carefully reviewing the author's modification, the manuscript has been greatly improved and can be considered for publication.